# Enhanced Photocatalytic Performance and Mechanism of Au@CaTiO_3_ Composites with Au Nanoparticles Assembled on CaTiO_3_ Nanocuboids

**DOI:** 10.3390/mi10040254

**Published:** 2019-04-17

**Authors:** Yuxiang Yan, Hua Yang, Zao Yi, Ruishan Li, Xiangxian Wang

**Affiliations:** 1State Key Laboratory of Advanced Processing and Recycling of Non-ferrous Metals, Lanzhou University of Technology, Lanzhou 730050, China; yanyx@lut.cn (Y.Y.); liruishan@lut.cn (R.L.); wangxx869@lut.edu.cn (X.W.); 2Joint Laboratory for Extreme Conditions Matter Properties, Southwest University of Science and Technology, Mianyang 621010, China; yizaomy@swust.edu.cn

**Keywords:** CaTiO_3_ nanocuboids, Au nanoparticles, localized surface plasmon resonance, Au@CaTiO_3_ composite, photocatalytic performance

## Abstract

Using P25 as the titanium source and based on a hydrothermal route, we have synthesized CaTiO_3_ nanocuboids (NCs) with the width of 0.3–0.5 μm and length of 0.8–1.1 μm, and systematically investigated their growth process. Au nanoparticles (NPs) of 3–7 nm in size were assembled on the surface of CaTiO_3_ NCs via a photocatalytic reduction method to achieve excellent Au@CaTiO_3_ composite photocatalysts. Various techniques were used to characterize the as-prepared samples, including X-ray powder diffraction (XRD), scanning/transmission electron microscopy (SEM/TEM), diffuse reflectance spectroscopy (UV-vis DRS), Fourier transform infrared spectroscopy (FTIR), and X-ray photoelectron spectroscopy (XPS). Rhodamine B (RhB) in aqueous solution was chosen as the model pollutant to assess the photocatalytic performance of the samples separately under simulated-sunlight, ultraviolet (UV) and visible-light irradiation. Under irradiation of all kinds of light sources, the Au@CaTiO_3_ composites, particularly the 4.3%Au@CaTiO_3_ composite, exhibit greatly enhanced photocatalytic performance when compared with bare CaTiO_3_ NCs. The main roles of Au NPs in the enhanced photocatalytic mechanism of the Au@CaTiO_3_ composites manifest in the following aspects: (1) Au NPs act as excellent electron sinks to capture the photoexcited electrons in CaTiO_3_, thus leading to an efficient separation of photoexcited electron/hole pairs in CaTiO_3_; (2) the electromagnetic field caused by localized surface plasmon resonance (LSPR) of Au NPs could facilitate the generation and separation of electron/hole pairs in CaTiO_3_; and (3) the LSPR-induced electrons in Au NPs could take part in the photocatalytic reactions.

## 1. Introduction

The rapid social development has raised two big issues facing mankind, i.e., environmental pollution and energy shortage. In particular, water resources are seriously and increasingly becoming polluted by the wastewater discharged from chemical industries, which poses a great threat to human health and survival. Organic dyes and pigments, most commonly existing in the industrial wastewater, are carcinogenic to humans and hardly self-decomposed [1,2]. It is imperative to remove the organic pollutants and purify water resources via a simple, low-cost, green and non-fossil-consumptive technology. In this sense, semiconductor-based photocatalysis has sparked a great interest due to its potential applications in wastewater treatment [3,4,5,6,7,8]. This technology allows the use of solar light—a sustainable, inexhaustible and economically attractive energy source—as the power source to degrade organic pollutants. The photocatalytic process depends highly on photogenerated electrons (e^−^) and holes (h^+^) in semiconductor photocatalysts, as well as their reduction and oxidation capabilities. Nevertheless, the photoexcited e^−^/h^+^ pairs are easily to be recombined and only a few of them are available for the photocatalytic reactions. To achieve an excellent semiconductor photocatalyst, the photoexcited e^−^/h^+^ pairs must be efficiently separated. As an important class of photocatalysts, titanium-contained oxide semiconductors can be photocatalytically active only under ultraviolet (UV) irradiation owing to their wide bandgap (*E*_g_ = 3.1–3.3 eV) [9,10,11,12,13]. It is noted that solar radiation includes only a small portion of UV light (~5%), but a large amount of visible light (45%). Enhancing the visible-light absorption of photocatalysts is the key point to make the best use of solar energy to drive the photocatalysis. Up to now, various strategies have been widely applied to modify semiconductor photocatalysts with the aim of facilitating the photoexcited e^−^/h^+^ pair separation and widening their light absorption range [14,15,16,17,18,19,20].

Zero-dimensional, one-dimensional and two-dimensional nanomaterials (e.g., metal nanoparticles (NPs), carbon quantum dots, carbon nanotubes, graphene) have attracted a great deal of interest due to their interesting physicochemical characteristics and great potential applications in the fields of bioimaging, energy conversion, optoelectronic devices, wave absorption and sensors [21,22,23,24,25,26,27,28,29,30,31,32,33]. Furthermore, these nanomaterials can be used as excellent modifiers or co-catalysts and are widely coupled with semiconductors to improve their photocatalytic performances [34,35,36,37,38]. Noble metal NPs are particularly interesting as the co-catalysts because they can not only facilitate the photoexcited e^−^/h^+^ pair separation but also enhance visible-light absorption. The enhanced photocatalytic mechanisms by noble metal NPs can be ascribed to two aspects [39,40]. First, noble metal NPs can act as electron sinks to trap photogenerated electrons from the semiconductor, thus leading to an efficient separation of e^−^/h^+^ pairs. Second, noble metal NPs can absorb visible light to induce localized surface plasmon resonance (LSPR) [41,42]. The LSPR-caused electromagnetic field could facilitate the generation and separation of e^−^/h^+^ pairs in the semiconductor. Simultaneously, LSPR-induced electrons in noble metal NPs could also take part in the photocatalytic reactions. Due to these unique properties, much work has demonstrated that noble metal NPs decorated semiconductors manifest significantly enhanced photocatalytic performances when compared with bare semiconductors [15,38,39,40].

Calcium titanate (CaTiO_3_), a typical titanium-contained oxide semiconductor with a perovskite-type structure, has sparked a great interest among researchers owing to its promising properties of ferroelectricity, piezoelectricity, elasticity, and photocatalytic activity [43,44,45,46,47]. As a photocatalyst, CaTiO_3_ has been shown to exhibit a pronounced photocatalytic degradation of organic pollutants, as well as photocatalytic splitting of water into hydrogen and oxygen [48,49,50,51,52]. Semiconductor-based photocatalysis, intrinsically being a heterogeneous surface catalytic reaction, is highly dependent on the crystal morphology of the semiconductor. In particular, an excellent photocatalytic activity could be achieved for the semiconductor with special exposed facets [53,54]. In our previous studies, we have demonstrated that CaTiO_3_ nanocuboids (NCs) with (101) and (010) exposed facets exhibit a photocatalytic activity superior to that of sphere-like CaTiO_3_ nanoparticles [55]. In this work, we have assembled Au NPs on the surface of CaTiO_3_ NCs and found that the obtained Au@CaTiO_3_ composites exhibit much enhanced photocatalytic degradation of organic dyes under both UV and visible light irradiation. Compared to Ag NPs, Au NPs offer an advantage of higher chemical stability. However, there is no work concerned with the Au NPs modified CaTiO_3_ photocatalysts, though Ag NPs have been frequently used as a co-catalyst to improve the photocatalytic performance of CaTiO_3_ [56,57,58]. Here we also systematically investigated the growth process of CaTiO_3_ photocatalysts. The photocatalytic mechanism of the Au@CaTiO_3_ composites were systematically investigated and discussed. The present Au@CaTiO_3_ nanocomposite photocatalysts can be introduced in micro/nano photocatalytic devices for the wastewater treatment.

## 2. Materials and Methods

### 2.1. Synthesis of CaTiO_3_ NCs

CaTiO_3_ NCs were synthesized via a hydrothermal route at 200 °C as described in the literature [55]. To unveil the growth process of CaTiO_3_ NCs, different hydrothermal reaction times (0.5, 1, 5, 10 and 24 h) were performed.

### 2.2. Assembly of Au NPs on CaTiO_3_ NCs

A photocatalytic reduction method was employed to hybridize Au NPs on the surface of CaTiO_3_ NCs synthesized at 200 °C for 24 h. 0.1 g of the as-synthesized CaTiO_3_ NCs and 0.025 g of ammonium oxalate (AO) were successively added in 80 mL of deionized water. The suspension was ultrasonically treated for 30 min and then magnetically stirred for another 30 min to make CaTiO_3_ particles uniformly disperse. 0.8 mL of HAuCl_4_ solution (0.029 mol·L^−1^, M) was dropped in the CaTiO_3_ suspension. The resultant mixture was magnetically stirred for 60 min, and then irradiated with ultraviolet (UV) light, which was emitted from a 15 W low-pressure mercury lamp, for 30 min under mild stirring. During the irradiation process, Au^3+^ ions were reduced by the photogenerated electrons in CaTiO_3_ to form Au NPs, which were simultaneously assembled on the surface of CaTiO_3_ NCs. The product was collected by centrifugation. After washing several times with deionized water and ethanol, and then drying at 60 °C for 12 h, the final product was obtained as the 4.3%Au@CaTiO_3_ composite with an Au mass fraction of 4.3%. By adding different volumes of HAuCl_4_ solution (0.2, 0.5, 1.1 and 1.4 mL), several other composite samples 1.1%Au@CaTiO_3_, 2.7%Au@CaTiO_3_, 5.9%Au@CaTiO_3_ and 7.4%Au@CaTiO_3_ were also prepared.

### 2.3. Sample Characterization

The crystal structures, morphologies and microstructures of the samples were characterized by X-ray powder diffraction (XRD) and scanning/transmission electron microscopy (SEM/TEM). The used apparatuses were a D8 Advance X-ray diffractometer (Bruker AXS, Karlsruhe, Germany), a JSM-6701F field-emission scanning electron microscope (JEOL Ltd., Tokyo, Japan) and a JEM-1200EX field-emission transmission electron microscope (JEOL Ltd., Tokyo, Japan). A PHI-5702 multi-functional X-ray photoelectron spectrometer (Physical Electronics, hanhassen, MN, USA) was employed for the X-ray photoelectron spectroscopy (XPS) analysis. Ultraviolet-visible (UV-vis) diffuse reflectance spectroscopy (DRS) measurements were performed on a TU-1901 double beam UV-vis spectrophotometer (Beijing Purkinje General Instrument Co. Ltd., Beijing, China). A Spectrum Two Fourier transform infrared (FTIR) spectrophotometer (PerkinElmer, Waltham, MA, USA) was used for the FTIR spectroscopy analysis of the samples.

### 2.4. Photocatalytic Test

To assess the photocatalytic performances of the samples, RhB in aqueous solution (5 mg·L^−1^) was used as the model pollutant. Simulated sunlight (emitted from a 200 W xenon lamp, 300 < λ < 2500 nm), UV light (emitted from a 30 W low-pressure mercury lamp, λ = 254 nm) and visible light (generated by a 200 W halogen-tungsten lamp, λ > 400 nm) were separately used as the light source. The reaction solution was composed of 100 mL of RhB solution and 0.1 g of the photocatalyst. The adsorption of RhB on the photocatalyst surface was examined by magnetically stirring the mixture in the dark for 30 min. The RhB concentration was monitored by measuring the absorbance of the reaction solution. 2.5 mL of the reaction solution was sampled from the photoreactor and centrifuged to remove the photocatalyst. The absorbance measurement was performed on a UV-vis spectrophotometer at λ = 554 nm. The percentage degradation of RhB (*D*%) was obtained according to *D*% = (*C*_0_ − *C*_t_)/*C*_0_ × 100% (*C*_0_ = initial RhB concentration; *C*_t_ = residual RhB concentration).

## 3. Results and Discussion

### 3.1. Synthesis and Growth Process of CaTiO_3_ NCs

The crystal structures of precursor P25 and the samples prepared at 200 °C with different reaction times (0.5, 1, 5, 10 and 24 h) were determined by XRD patterns, as shown in Figure 1. It is seen that at 0.5 h reaction time, the obtained sample maintains a crystal structure identical to that of P25 without the formation of CaTiO_3_ phase. At 1 h reaction time, CaTiO_3_ phase is largely crystallized and only minor TiO_2_ is observed in the resultant sample. When the reaction time is increased up to 5 h, single CaTiO_3_ phase is obtained. All the diffraction peaks of the sample can be indexed to the CaTiO_3_ orthorhombic phase (JCPDS#42-0423). With further prolonging the reaction time, the diffraction peaks of the resultant samples undergo no change, indicating no structural change of CaTiO_3_ crystals.

Figure 2 illustrates the SEM images of precursor P25 and the samples prepared at different reaction times. It is seen that P25 is composed of sphere-like nanoparticles with average size of 25 nm (Figure 2a). At 0.5 h of reaction, the TiO_2_ nanoparticles undergo almost no change, as shown in Figure 2b. The sample derived at 1 h reaction time mainly consists of cuboid-like particles together with minor sphere-like nanoparticles (Figure 2c), indicating that most of TiO_2_ nanoparticles are coupled with Ca species to form CaTiO_3_ NCs. With increasing the reaction time up to 5 h, TiO_2_ nanoparticles disappear and single CaTiO_3_ NCs are formed, as depicted in Figure 2d. These CaTiO_3_ NCs have a size of 0.3–0.5 μm in width and 0.8–1.1 μm in length. Further prolonging the reaction time up to 10 h (Figure 2e) and 24 h (Figure 2f) leads to no obvious morphological change of CaTiO_3_ NCs.

The growth process and mechanism of CaTiO_3_ NCs is schematically illustrated in Figure 3. In the concentrated NaOH solution, polycrystalline TiO_2_ nanoparticles are expected to react with NaOH to form Na_2_Ti_2_O_4_(OH)_2_ particles [59]. Simultaneously, in the strong alkaline environment and high temperature-high pressure conditions, Na_2_Ti_2_O_4_(OH)_2_ dissociates to form Ti(OH)_6_^2−^ ion groups [60]. Ca^2+^ ions in the precursor solution are adsorbed onto the surface of Ti(OH)_6_^2−^ ions and further penetrate into the interior. During this process, a series of chemical reactions will take place, including the breaking of Ti-O-Ti bonds, dehydroxylation and nucleation of CaTiO_3_ crystals. To reduce the overall surface energy, the nucleated CaTiO_3_ crystals grow into nanocuboids finally. The dominant chemical reactions involved can be briefly described as follows:2NaOH + 2TiO_2_ → Na_2_Ti_2_O_4_(OH)_2_(1)
Na_2_Ti_2_O_4_(OH)_2_ + 2OH^−^ + 4H_2_O → 2Ti(OH)_6_^2−^ + 2Na^+^(2)
Ca^2+^ + Ti(OH)_6_^2−^ → CaTiO_3_ + 3H_2_O(3)

### 3.2. Au NPs Modified CaTiO_3_ NCs

CaTiO_3_ NCs synthesized at 200 °C for 24 h was chosen to be modified with Au NPs with the aim of enhancing their photocatalytic performance. Figure 4 shows the XRD patterns of bare CaTiO_3_ NCs and 4.3%Au@CaTiO_3_ composite. The dominant diffraction peaks of 4.3%Au@CaTiO_3_ are similar to those of bare CaTiO_3_, indicating that the CaTiO_3_ orthorhombic structure undergoes no change on the decoration of Au NPs. However, additional weak diffraction peaks characterized as the Au cubic structure (JCPDS#04-0784) are clearly detected on the XRD pattern of the composite, which confirms the formation of Au NPs onto CaTiO_3_ NCs.

TEM investigation was further carried out to unveil the microstructure of the 4.3%Au@CaTiO_3_ composite. Figure 5a shows the TEM image of the composite, from which one can see that CaTiO_3_ presents a regular morphology of NCs with width of 0.3–0.5 μm and length of 0.8–1.1 μm. The CaTiO_3_ morphology obtained by TEM is in accordance with the SEM observation result. Moreover, small-sized Au NPs are seen to be decorated on the surface of CaTiO_3_ NCs. The high-resolution TEM (HRTEM) image depicted in Figure 5b further confirms the good assembly of Au NPs on the surface of CaTiO_3_ NCs. The decorated Au NPs are shaped like spheres and have a size distribution of 3–7 nm. Energy-dispersive x-ray elemental mapping was used to investigate the elemental distribution of the 4.3%Au@CaTiO_3_ composite. Figure 5c illustrates the dark-field scanning TEM (DF-STEM) image of the composite, and the corresponding elemental mapping images are given in Figure 5d–g. It is observed that the NCs present uniformly-distributed elements of Ca, Ti and O. Moreover, Au element is also seen to be uniformly distributed throughout the NCs, indicating that CaTiO_3_ NCs are uniformly decorated with Au NPs. Energy-dispersive X-ray spectroscopy (EDS) spectrum was further used to examine the chemical composition of the 4.3%Au@CaTiO_3_ composite. As shown in Figure 5h, the composite sample is composed of Ca, Ti, O, and Au elements. Additional Cu and C signals detected on the EDS spectrum could come from the TEM microgrid holder [61]. The obtained Au content from the EDS spectrum is 4.1%, which is basically in agreement with the stoichiometric composition of the 4.3%Au@CaTiO_3_ composite.

The XPS analyses were performed on 4.3%Au@CaTiO_3_ to elucidate the chemical states of elements. Figure 6a shows the survey XPS spectrum, revealing that the composite is composed of the elements Ca, Ti, O and Au. The observed C signal comes from adventitious carbon, which is used for the calibration of binding energy (C 1s binding energy: 284.8 eV). On the high-resolution XPS spectrum of Ca 2p core level (Figure 6b), the observed peaks at 346.7 and 350.2 eV account for Ca 2p_3/2_ and Ca 2p_1/2_, respectively. On the Ti 2p XPS spectrum (Figure 6c), the Ti 2p_3/2_ and Ti 2p_1/2_ binding energies are observed at 458.8 and 464.4 eV, respectively. The binding energy positions suggest that the Ca and Ti species behave as Ca^2+^ and Ti^4+^ oxidation states, respectively [13]. Two peaks at 529.9 and 532.1 eV are detected on the O 1s XPS spectrum (Figure 6d), which are assigned to the crystal lattice oxygen in CaTiO_3_ and chemisorbed oxygen species [13,62]. On the Au 4f XPS spectrum (Figure 6e), the peak at 83.6 is attributed to Au 4f_7/2_ and the peak at 87.2 eV corresponds to Au 4f_5/2_. This implies the Au species exists in the metallic state [63].

Figure 7a,b depict the UV-vis DRS spectra of CaTiO_3_ and 4.3%Au@CaTiO_3_, respectively, along with the corresponding first derivative curves of the UV-vis DRS spectra and the digital images of the samples (insets). It is seen that the decoration of Au NPs onto CaTiO_3_ NCs obviously enhances the visible-light absorption. This is further confirmed by the deepening of the apparent color for the 4.3%Au@CaTiO_3_ composite (dark gray), as compared to bare CaTiO_3_ NCs (cream white). The enhanced visible-light absorption implies that the Au@CaTiO_3_ composite photocatalyst can utilize photons more effectively. From the first derivative spectra, the absorption edge of CaTiO_3_ and 4.3%Au@CaTiO_3_ is obtained as 369.3 and 355.0 nm, respectively [64]. This suggests that bare CaTiO_3_ has a bandgap energy (*E*_g_) of 3.36 eV and 4.3%Au@CaTiO_3_ exhibits an *E*_g_ of 3.49 eV. The slight increase in the *E*_g_ of the composite could be due to the interaction between CaTiO_3_ NCs and Au NPs.

We measured the FTIR spectra of CaTiO_3_ and 4.3%Au@CaTiO_3_ to elucidate their functional groups, as shown in Figure 8. The absorption peaks at 437 and 560 cm^−1^ are characterized as the Ti–O stretching vibration and Ti–O–Ti bridging stretching mode [65,66]. This implies the existence of TiO_6_ octahedra and the formation of CaTiO_3_ perovskite-type structure. The observed broad bands at 3430 (H_2_O stretching vibration) and 1637 cm^−1^ (H_2_O bending vibration) suggest the presence of water molecules absorbed on the surface of the samples [67]. The presence of NH^3+^ group can be confirmed by the N–H stretching vibration located at around 3146 cm^−1^ [68]. The peaks detected at 1395 and 1102 cm^−1^ are attributed to the O–H in-plane deformation and C–OH stretching vibrations of alcohols [34]. This indicates that alcohols could be anchored on the samples during their washing process. In addition, no characteristic peaks assignable to Au oxides are observed for the 4.3%Au@CaTiO_3_ composite, implying Au species exists in the metallic state.

To assess the photocatalytic degradation of RhB over the samples, three types of light source were separately used, i.e., simulated sunlight (300 < λ < 2500 nm), UV light (λ = 254 nm) and visible light (λ > 400 nm). Figure 9a shows the simulated-sunlight degradation of RhB photocatalyzed by CaTiO_3_ and Au@CaTiO_3_ composites. It is demonstrated that the Au@CaTiO_3_ composites manifests a photocatalytic activity much superior to that of bare CaTiO_3_ NCs. The content of decorated Au NPs has an obvious effect on the photocatalytic activity of the composite samples. With increasing the Au content, the photocatalytic activity of the samples gradually increases and reaches the highest level for 4.3%Au@CaTiO_3_. However, further increasing the Au content gives rise to a decrease in the photocatalytic activity. This could be explained by the fact that excessive decoration of Au NPs on the surface of CaTiO_3_ NCs could reduce the light absorption of CaTiO_3_. The degradation percentage of RhB after 120 min of photocatalytic reaction is inserted in Figure 9a. For the optimal composite sample—4.3%Au@CaTiO_3_, the dye degradation reaches 99.6%, which is increased by 23.2% when compared with that for bare CaTiO_3_ (76.4%). We also carried out the kinetic analysis of the dye degradation over the samples, as illustrated in Figure 9b. One can see that the plots of Ln(C_t_/C_0_) vs. *t* present a good linear relationship and can be perfectly described using the pseudo-first-order kinetic equation: Ln(*C*_t_/*C*_0_) = −*k*_app_*t* [69,70]. Based on the linear-regression fitting, the apparent first-order reaction rate constant *k*_app_ is obtained, as inserted in Figure 9b. It is concluded from the reaction rate constants (*k*_app_(CaTiO_3_) = 0.01195 min^−1^; *k*_app_(4.3%Au@CaTiO_3_) = 0.04701 min^−1^) that the photocatalytic activity of 4.3%Au@CaTiO_3_ is ~3.9 times as large as that of bare CaTiO_3_. Further comparison of the UV and visible-light photocatalytic performance between CaTiO_3_ and 4.3%Au@CaTiO_3_ was carried out. As shown in Figure 9c, CaTiO_3_ exhibits an important UV photocatalytic activity toward the degradation of RhB. Moreover, a significantly enhanced UV photocatalytic activity is observed for the 4.3%Au@CaTiO_3_ composite photocatalyst, which is ca. 3.0 times larger than that of bare CaTiO_3_. The visible-light photocatalytic degradation of RhB over CaTiO_3_ and 4.3%Au@CaTiO_3_ is shown in Figure 9d. It is observed that bare CaTiO_3_ photocatalyzes only 15.1% degradation of the dye after 120 min of photocatalysis, which could be ascribed to the dye-photosensitized degradation. This is indicative of a poor photocatalytic activity of CaTiO_3_. Whereas 46.1% of RhB is observed to be degraded over 4.3%Au@CaTiO_3_, implying a greatly enhanced visible-light photocatalytic activity of the composite.

Based on the above experimental results, we propose a possible mechanism to elucidate the enhanced photocatalytic performance of the Au NPs modified CaTiO_3_ NCs, as schematically depicted Figure 10. Under UV irradiation, CaTiO_3_ is excited to produce electrons in the CB and holes in the valence band (VB) of the semiconductor. Au NPs cannot be excited under UV irradiation, but they can act as excellent electron sinks to capture the photogenerated electrons in CaTiO_3_. This is because the Fermi level of Au (+0.45 V vs. normal hydrogen electrode (NHE) [71,72]) is more positive than the CB potential of CaTiO_3_ (−0.43 V vs. NHE [55]). The electron transfer process from the CB of CaTiO_3_ to Au NPs promotes the spatial separation of the electron/hole pairs in CaTiO_3_. This is supported by photoluminescence spectra (Appendix A), photocurrent response curves (Appendix A) and electrochemical impedance spectroscopy (EIS) spectra (Appendix A). As a result, more holes in the VB of CaTiO_3_ are available for taking part in the photocatalytic reactions. Under visible-light irradiation, CaTiO_3_ cannot be directly excited to produce electron/hole pairs, instead LSPR of Au NPs is induced by the visible-light absorption [39,40]. The LSPR-induced electrons in Au NPs could take part in the photocatalytic reactions, and moreover, the electromagnetic field caused by LSPR could stimulate the generation of electron/hole pairs in CaTiO_3_. This is why the Au@CaTiO_3_ composites also manifest enhanced visible-light photocatalytic degradation of RhB. When simulated sunlight is used as the light source, Au NPs simultaneously act as electron sinks and behave as LSPR effect in the Au@CaTiO_3_ composites. The two mechanisms collectively result in the enhanced photocatalytic performance of the composites under simulated sunlight irradiation.

Hydroxyl (•OH), superoxide (•O_2_^−^) and h^+^ are generally considered to be the main active species in most of photocatalytic systems [73]. The reactive species trapping experiments (Appendix A) reveal that the degradation of RhB is highly correlated with •OH and h^+^. However, the photogenerated h^+^ could not directly oxide the dye, but rather reacts with OH^−^ or H_2_O to produce another stronger oxidant •OH. Thermodynamically •OH is ready to be generated through the reaction between h^+^ and OH^−^/H_2_O due to the sufficiently positive VB potential of CaTiO_3_ (+2.93 V vs. NHE [55]) when compared with the redox potentials of H_2_O/•OH (+2.38 V vs. NHE) and OH^−^/•OH (+1.99 V vs. NHE) [74]. •O_2_^−^ plays only a slight role in the photocatalytic degradation of the dye. The sufficiently negative CB potential of CaTiO_3_ (−0.43 V vs. NHE) suggests that the generation of •O_2_^−^ can be derived from the reaction of adsorbed O_2_ with the CB electrons of CaTiO_3_ (*E*^0^(O_2_/•O_2_^−^ = −0.13 V vs. NHE) [74]. Moreover, the LSPR-induced electrons in Au NPs could also combine with O_2_ to produce •O_2_^−^. Recycling photocatalytic experiment (Appendix A) is indicative of a good recycling stability of the Au@CaTiO_3_ composite photocatalyst.

## 4. Conclusions

Based on a hydrothermal route, CaTiO_3_ NCs were synthesized using P25 as the titanium source, and their growth process was investigated by varying the reaction time. Au NPs were uniformly decorated on the CaTiO_3_ NCs surface by a photocatalytic reduction of HAuCl_4_ solution. Compared to bare CaTiO_3_ NCs, the as-derived Au@CaTiO_3_ composites manifest an increased visible-light absorption, increased photocurrent density, decreased charge-transfer resistance, decreased PL intensity, as well as enhanced photocatalytic performance for the degradation of RhB under irradiation of different light sources (simulated sunlight, UV light and visible light). The optimal composite sample, observed to be 4.3%Au@CaTiO_3_, has a photocatalytic activity 3.9 times as large as that of bare CaTiO_3_ NCs, and photocatalyzes 99.6% degradation of RhB under simulated sunlight irradiation for 120 min. The enhanced photocatalytic mechanism of the Au@CaTiO_3_ composites can be explained by (1) the efficient separation of photoexcited e^−^/h^+^ pairs in CaTiO_3_ due to the electron transfer from CaTiO_3_ NCs to Au NPs, and (2) increased visible-light absorption due to the LSPR effect of Au NPs. The LSPR-caused electromagnetic field could stimulate the generation and separation of e^−^/h^+^ pairs in CaTiO_3_, and moreover, the LSPR-induced electrons in Au NPs could be available for the photodegradation reactions. Reactive species trapping experiments reveal that the dye degradation is highly correlated with •OH radicals and photoexcited holes, whereas •O_2_^−^ radicals play only a slight role in the photocatalysis. The present Au@CaTiO_3_ nanocomposite photocatalysts could offer promising applications in the design of micro/nano photocatalytic devices for the wastewater treatment.

## Figures and Tables

**Figure 1 micromachines-10-00254-f001:**
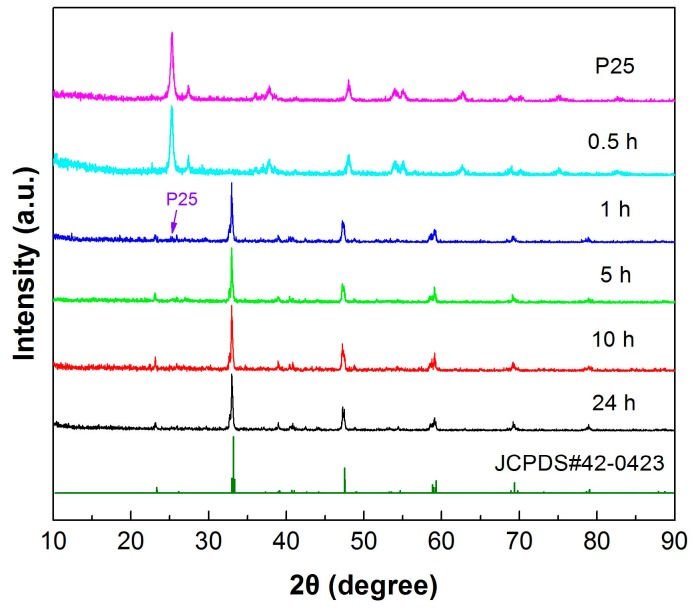
X-ray powder diffraction (XRD) patterns of precursor P25 and the samples prepared at 200 °C with different reaction times (0.5, 1, 5, 10 and 24 h).

**Figure 2 micromachines-10-00254-f002:**
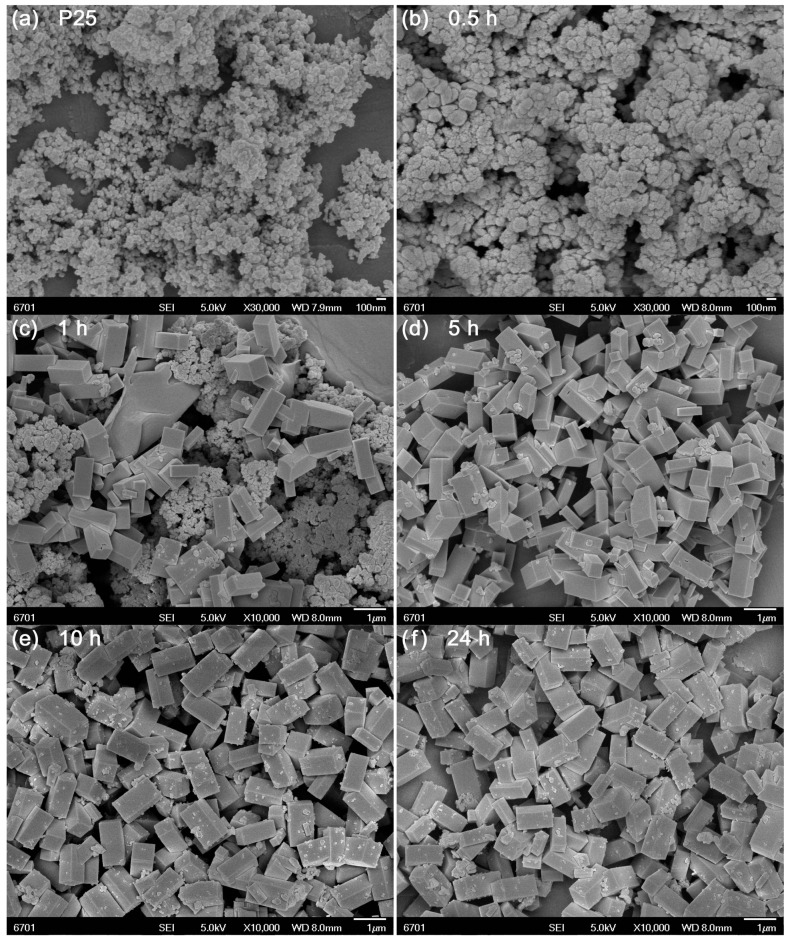
Scanning electron microscopy (SEM) images of (**a**) precursor P25 and the samples prepared at (**b**) 0.5, (**c**) 1, (**d**) 5, (**e**) 10 and (**f**) 24 h.

**Figure 3 micromachines-10-00254-f003:**
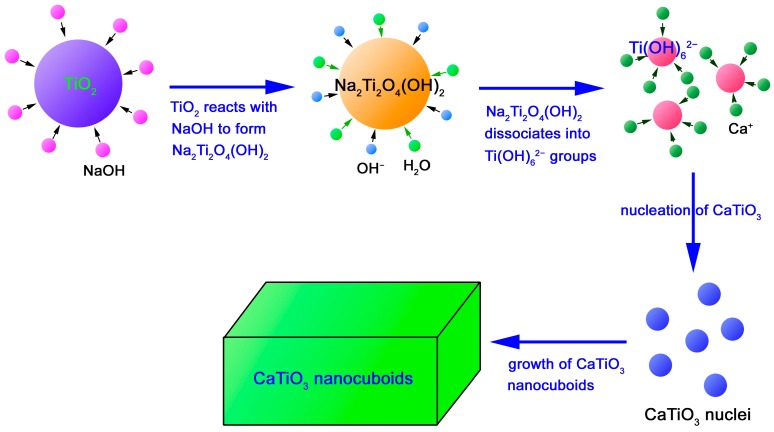
Schematic illustration of the growth process and mechanism of CaTiO_3_ NCs.

**Figure 4 micromachines-10-00254-f004:**
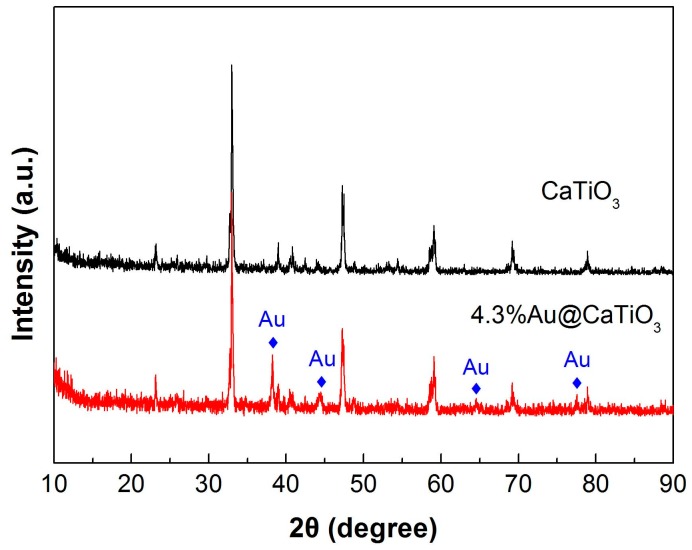
XRD patterns of bare CaTiO_3_ NCs and the 4.3%Au@CaTiO_3_ composite.

**Figure 5 micromachines-10-00254-f005:**
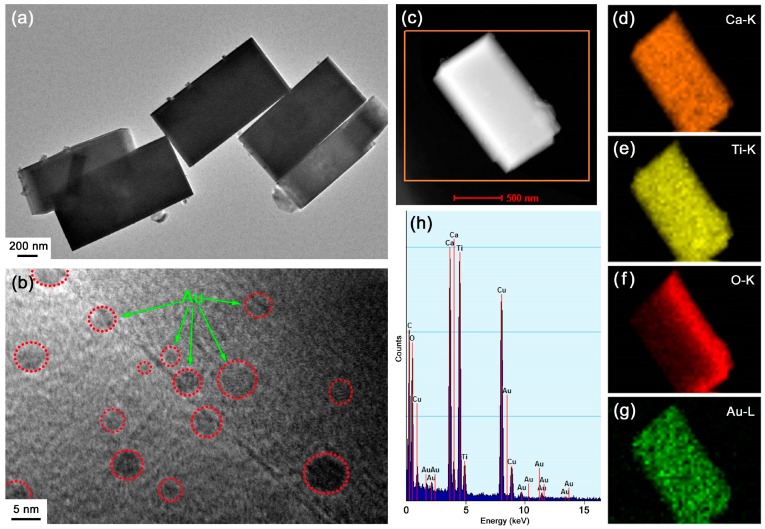
Transmission electron microscopy (TEM) image (**a**), HRTEM image (**b**), DF-STEM image (**c**), elemental mapping images (**d**–**g**), and EDS spectrum (**h**) of the 4.3%Au@CaTiO_3_ composite.

**Figure 6 micromachines-10-00254-f006:**
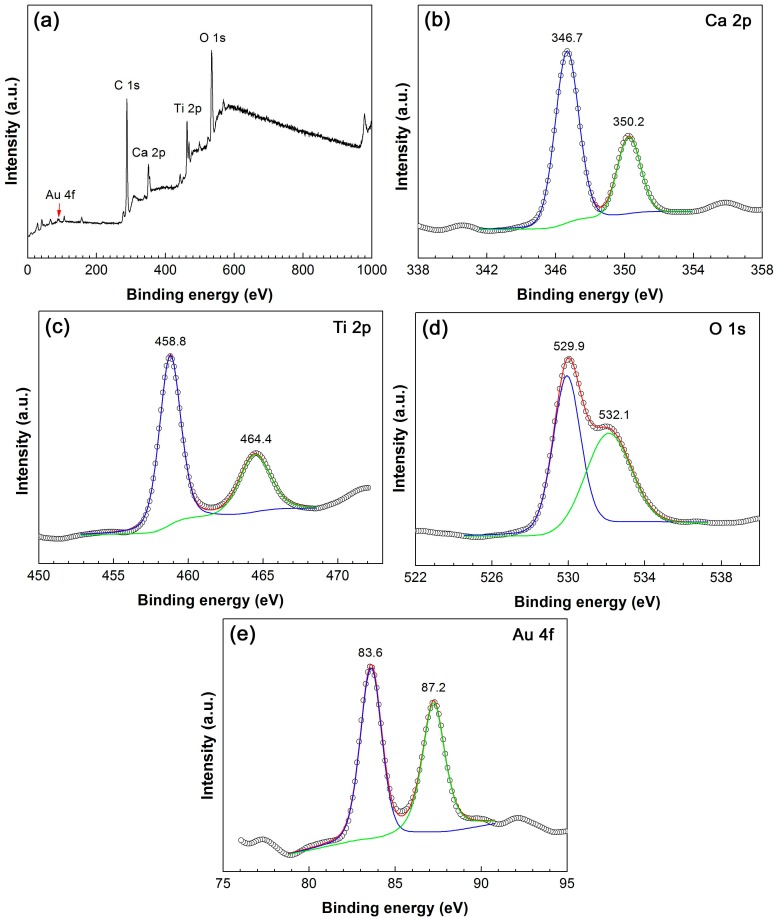
XPS survey scan spectrum (**a**), and high-resolution XPS spectra of (**b**) Ca 2p, (**c**) Ti 2p, (**d**) O 1s and (**e**) Au 4f of the 4.3%Au@CaTiO_3_ composite.

**Figure 7 micromachines-10-00254-f007:**
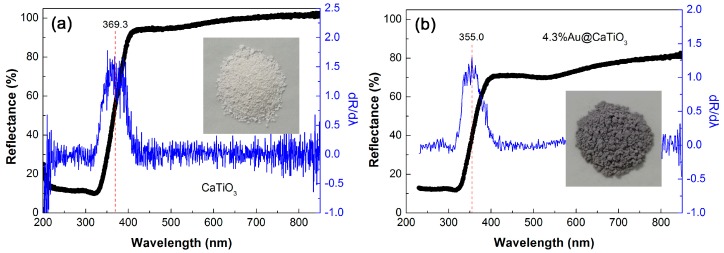
UV-vis diffuse reflectance spectroscopy (DRS) spectra, first derivative curves of the UV-vis DRS spectra and digital images (insets) of (**a**) CaTiO_3_ and (**b**) 4.3%Au@CaTiO_3_.

**Figure 8 micromachines-10-00254-f008:**
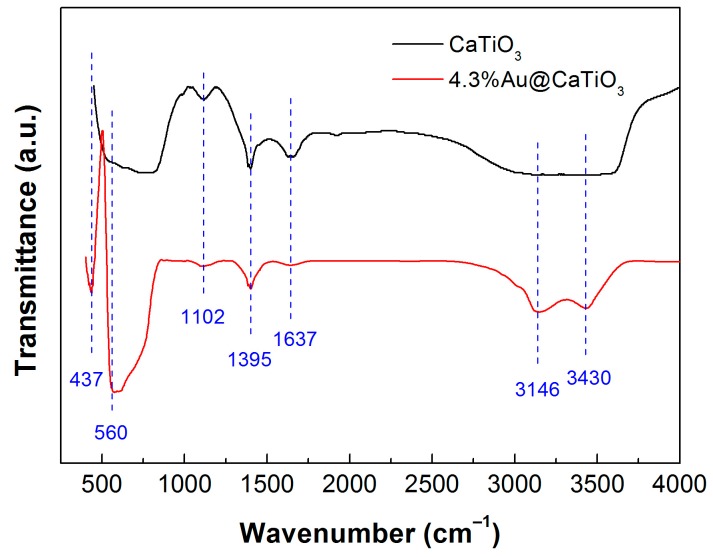
Fourier transform infrared spectroscopy (FTIR) spectra of CaTiO_3_ and 4.3%Au@CaTiO_3_.

**Figure 9 micromachines-10-00254-f009:**
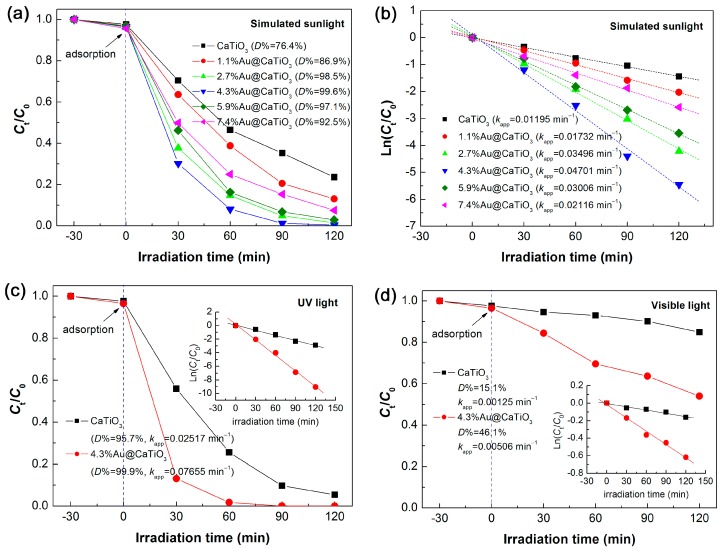
(**a**) Time-dependent photocatalytic degradation of RhB over bare CaTiO_3_ and Au@CaTiO_3_ composites under simulated sunlight irradiation. (**b**) Kinetic plots of the dye degradation over the samples under simulated sunlight irradiation. (**c**) UV and (**d**) visible-light photocatalytic degradation of RhB over bare CaTiO_3_ and 4.3%Au@CaTiO_3_.

**Figure 10 micromachines-10-00254-f010:**
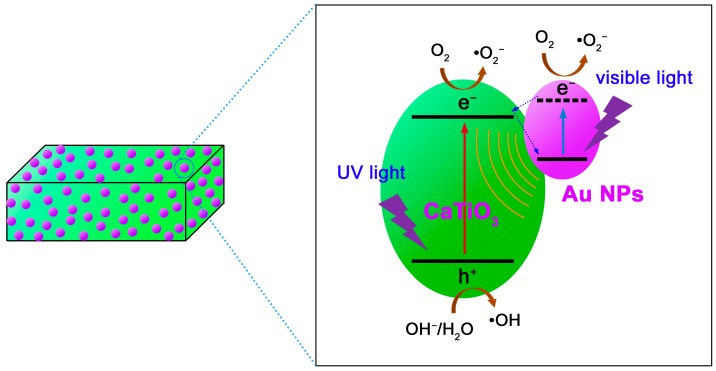
Schematic illustration of the photocatalytic mechanism of the Au@CaTiO_3_ composites.

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
