# Peer review of "Enhanced Photocatalytic Performance and Mechanism of Au@CaTiO3 Composites with Au Nanoparticles Assembled on CaTiO3 Nanocuboids"

_micromachines, 2019, doi:10.3390/mi10040254_

Round 1
Reviewer 1 Report
This is a well written manuscript. Overall this adds to the environmental remediation applications and the study is novel with enhanced photocatalytic activity.
Some of the latest relevant article (Nanotechnology applications for environmental industry Handbook of nanomaterials for industrial applications, 894-907), can be included in the introduction section.
Author Response
Some of the latest relevant article (Nanotechnology applications for environmental industry Handbook of nanomaterials for industrial applications, 894-907), can be included in the introduction section.
Response: We thank very much the Reviewer for reviewing our manuscript and giving a positive comment. We cited the article mentioned by the Reviewer.

Reviewer 2 Report
This manuscript describes the generation of hybrid materials (Au NP on CaTiO3) as improved photocatalysts for wastewater cleaning. The topic is important, and the synthesis and characterization appear to have been carefully performed, but it is somewhat difficult to evaluate the extent to which the results are high-impact based solely on the information provided.
In the methods, both 2.1 and 2.2 could use additional citations that discuss how these synthesis approaches relate to that done by others. As described in greater detail below, some rewriting could improve the clarity of the manuscript, particularly where the mechanisms are concerned. The current manuscript contains many figures that could be relocated to supplemental information, as could the detailed discussion of said figures. In particular, the PL spectra (can the lifetime say something about the relative materials?), photocurrent, and Nyquist plots conrtibute only minor evidence in support of their arguments, and, while they should remain in the supplemental, the reader can become lost when trying to navigate the somewhat long characterization section. I advise the authors to carefully consider what aspects of Figure 11 must be in the main paper, and what could be moved, particularly because e and f are not as closely related to one another or to a-d.
The most useful organization change might be to wait to describe the hole, hydroxide, and superoxide scavenging events until after a clear set of hypotheses have been established. As I was reading, at line 290-293, the first assertions were made, but then the experiments came far after (two pages), and were followed by the scavenger analysis. Re-evaluate the order in which these are presented so that the reader is provided with a hypothesis, followed by an experiment, then results that confirm or reject the hypothesis.
I did not find the detailed FTIR analysis particularly informative. Shouldn't solvent peaks be removable by drying, thereby eliminating ambiguity?
The use of k to claim that the authors have developed a superior material is too relative. To assess this properly, the rate constant must be scaled by photon intensity, which will be different for each of the sources used, and may vary wildly when another group reproduces the work for comparison. Comparison to other well-known and well-characterized photocatalysts will also help place the work in context.
Overall, I find that this work has merit and could become publishable if these minor changes are made.
Author Response
This manuscript describes the generation of hybrid materials (Au NP on CaTiO3) as improved photocatalysts for wastewater cleaning. The topic is important, and the synthesis and characterization appear to have been carefully performed, but it is somewhat difficult to evaluate the extent to which the results are high-impact based solely on the information provided.
In the methods, both 2.1 and 2.2 could use additional citations that discuss how these synthesis approaches relate to that done by others. As described in greater detail below, some rewriting could improve the clarity of the manuscript, particularly where the mechanisms are concerned. The current manuscript contains many figures that could be relocated to supplemental information, as could the detailed discussion of said figures. In particular, the PL spectra (can the lifetime say something about the relative materials?), photocurrent, and Nyquist plots conrtibute only minor evidence in support of their arguments, and, while they should remain in the supplemental, the reader can become lost when trying to navigate the somewhat long characterization section. I advise the authors to carefully consider what aspects of Figure 11 must be in the main paper, and what could be moved, particularly because e and f are not as closely related to one another or to a-d.
Response: We thank very much the Reviewer for reading our manuscript and giving valuable suggestions. In the methods, we simplified 2.1 section according to the Reviewer’s suggestion. We relocated the PL spectra (Fig. 9), photocurrent (Fig. 10a), and Nyquist plots (Fig. 10b) to supplemental information. We also relocated Fig. 11e and f to supplemental information.
The most useful organization change might be to wait to describe the hole, hydroxide, and superoxide scavenging events until after a clear set of hypotheses have been established. As I was reading, at line 290-293, the first assertions were made, but then the experiments came far after (two pages), and were followed by the scavenger analysis. Re-evaluate the order in which these are presented so that the reader is provided with a hypothesis, followed by an experiment, then results that confirm or reject the hypothesis.
Response: According to the Reviewer’s suggestion, we have made the organization change.
I did not find the detailed FTIR analysis particularly informative. Shouldn't solvent peaks be removable by drying, thereby eliminating ambiguity?
Response: The FTIR spectra suggest some functional groups on the samples. It is noted that all the samples were dried at 60oC for 12 h, but the functional groups still exist, which indicates the functional groups are anchored on the surface of the samples.
The use of k to claim that the authors have developed a superior material is too relative. To assess this properly, the rate constant must be scaled by photon intensity, which will be different for each of the sources used, and may vary wildly when another group reproduces the work for comparison. Comparison to other well-known and well-characterized photocatalysts will also help place the work in context.
Response: We thank very much the Reviewer for giving us this good comment. We completely agree with the Reviewer’s comment. Unfortunately, however, our lab has no instrument to measure the photon intensity. We hope the Reviewer can understand that we cannot make further revision regarding this comment.
Overall, I find that this work has merit and could become publishable if these minor changes are made.
